# Diagnostic Accuracy of Neutrophil Gelatinase-Associated Lipocalin in Peritoneal Effluent and Ascitic Fluid for Early Detection of Peritonitis: A Systematic Review and Meta-Analysis

**DOI:** 10.3390/medsci13030175

**Published:** 2025-09-04

**Authors:** Manuel Luis Prieto-Magallanes, José David González-Barajas, Violeta Aidee Camarena-Arteaga, Bladimir Díaz-Villavicencio, Juan Alberto Gómez-Fregoso, Ana María López-Yáñez, Ruth Rodríguez-Montaño, Judith Carolina De Arcos-Jiménez, Jaime Briseno-Ramírez

**Affiliations:** 1Hospital Civil de Oriente, Nuevo Periferico Oriente s/n, Tónala C.P. 45425, Jalisco, Mexico; mlprieto@hcg.gob.mx (M.L.P.-M.); jdgonzalez@hcg.gob.mx (J.D.G.-B.); 2025009@correo.opdhcg.net (V.A.C.-A.); 2024673@correo.opdhcg.net (B.D.-V.); ana.lopez@academicos.udg.mx (A.M.L.-Y.); 2Antiguo Hospital Civil de Guadalajara “Fray Antonio Alcalde” Hospital 278, Guadalajara C.P. 44280, Jalisco, Mexico; juanalberto.gomez@academicos.udg.mx; 3Centro Universitario de Tlajomulco, Universidad de Guadalajara, Carretera Tlajomulco–Santa Fé Km. 3.5 No. 595, Tlajomulco de Zúñiga C.P. 45641, Jalisco, Mexico; ruth.rodriguez@academicos.udg.mx

**Keywords:** neutrophil gelatinase-associated lipocalin, peritonitis, diagnostic accuracy, meta-analysis, peritoneal dialysis, ascites

## Abstract

Background: Peritonitis in peritoneal dialysis and cirrhosis remains common and leads to morbidity. Neutrophil gelatinase-associated lipocalin (NGAL) has been evaluated as a rapid adjunctive biomarker. Methods: Following PRISMA-DTA and PROSPERO registration (CRD420251105563), we searched MEDLINE, Embase, Cochrane Library, LILACS, Scopus, and Web of Science from inception to 31 December 2024, and ran an update on 30 June 2025 (no additional eligible studies). Diagnostic accuracy studies measuring NGAL in peritoneal/ascitic fluid against guideline reference standards were included. When 2 × 2 data were not reported, we reconstructed cell counts from published metrics using a prespecified, tolerance-bounded algorithm (two studies). Accuracy was synthesized with a bivariate random effects (Reitsma) model; 95% prediction intervals (PIs) were used to express heterogeneity; small-study effects were assessed by Deeks’ test. Results: Thirteen studies were included qualitatively and ten were entered into a meta-analysis (573 cases; 833 controls). The pooled sensitivity was 0.95 (95% CI, 0.90–0.97) and specificity was 0.86 (0.70–0.94); likelihood ratios were LR+ ≈7.0 and LR− 0.06. Between-study variability was concentrated on specificity: the PI for a new setting was 0.75–0.98 for sensitivity and 0.23–0.99 for specificity. Deeks’ test showed evidence of small-study effects in the primary analysis; assay/platform and thresholding contributed materially to heterogeneity. Conclusions: NGAL in peritoneal/ascitic fluid demonstrates high pooled sensitivity but variable specificity across settings. Given the wide prediction intervals and the signal for small-study effects, NGAL should be interpreted as an adjunct to guideline-based criteria—not as a stand-alone rule-out test. Standardization of pre-analytics and assay-specific, locally verified thresholds, together with prospective multicenter validations and impact/economic evaluations, are needed to define its clinical role.

## 1. Introduction

Peritonitis represents a significant clinical burden in both dialysis patients and those with cirrhosis [1,2,3,4]. Among individuals receiving peritoneal dialysis (PD), peritoneal dialysis-associated peritonitis (PDAP) is the most frequent infectious complication [1]. Although incidence varies across programs, current benchmarks recommend rates not exceeding 0.5 episodes per patient-year, with many centers achieving ≤ 0.33 and some as low as 0.20 episodes per patient-year [1,5]. In the United States, recent data indicate an incidence of 0.26 episodes per patient-year; approximately 28% of patients experience at least one episode during the first PD year [5]. Nearly half of episodes require hospitalization and peritonitis-attributable mortality is 3–10%, with excess mortality persisting up to 120 days after an event [1,6,7,8].

Among patients with cirrhosis, spontaneous bacterial peritonitis (SBP) is the most prevalent and clinically consequential infection, accounting for roughly 27% of infections in this population [9]. Infections increase mortality fourfold, and in-hospital mortality with sepsis may exceed 50% [3,9,10]. The high prevalence of multidrug-resistant organisms, particularly in nosocomial settings, intensifies this burden [9,11]. In cirrhotic patients receiving PD, the risk of peritonitis is slightly higher than in non-cirrhotic PD patients (odds ratio 1.10; 95% CI 1.03–1.18), although overall mortality is not significantly increased [12].

Neutrophil gelatinase-associated lipocalin (NGAL) has emerged as a promising biomarker of peritoneal infection across these settings [13,14]. NGAL is released by neutrophils and mesothelial cells during infection-related inflammation, rises rapidly in peritoneal fluid, and participates in innate host defense by iron sequestration and modulation of neutrophil function [15,16,17,18].

In cirrhotic ascites, NGAL concentrations are higher in SBP than in non-SBP and correlate with the absolute neutrophil count. Diagnostic performance is generally good (AUC ~0.89; for example, sensitivity ~96% and specificity ~75% at ~120 ng/mL) and ascitic NGAL independently predicts in-hospital mortality [13,19,20,21]. In PD, NGAL in peritoneal dialysis effluent (PDE) increases early during PDAP, correlates with PDE white blood cell counts, may assist in distinguishing bacterial from non-bacterial/culture-negative peritonitis, and can decline earlier than WBC during recovery—features supporting a potential role in monitoring [15,22,23,24].

Despite these promising signals, clinical uptake has been limited by non-standardized assays and cut-offs, variable sampling/dwell conditions, and the predominance of small single-center or case-control studies; large prospective multicenter validations are scarce [13,20,22]. Accordingly, we synthesized evidence on NGAL in peritoneal dialysis effluent and ascitic fluid to provide robust, platform-aware estimates of diagnostic accuracy for PDAP and SBP and to clarify sources of variability that are most relevant for clinical implementation.

## 2. Materials and Methods

This systematic review and meta-analysis adhered to the PRISMA (Preferred Reporting Items for Systematic Reviews and Meta-Analyses) guidelines for diagnostic test accuracy (PRISMA-DTA) and used QUADAS 2 for risk of bias assessment [25,26]. It was prospectively registered in PROSPERO (CRD420251105563); the public record (Version 2.0, 31 July 2025) details the full protocol, search strategies, and amendments expanding the scope to include both PDAP and SBP. A scoping search preceded registration; no data extraction or quantitative synthesis was undertaken before registration.

### 2.1. Eligibility Criteria

We included studies in which human subjects of any age had undergone peritoneal fluid sampling—either dialysis effluent or ascitic fluid—to evaluate neutrophil gelatinase-associated lipocalin as a biomarker for peritonitis. For peritoneal dialysis-associated peritonitis, the reference standard was defined according to International Society for Peritoneal Dialysis (ISPD) criteria, requiring at least two of the following: clinical features consistent with peritonitis (abdominal pain and/or cloudy dialysis effluent), dialysis effluent WBC count > 100 cells/µL or >0.1 × 10^9^ L^−1^ with >50% polymorphonuclear leukocytes after a dwell time of at least two hours, and positive dialysis effluent culture [27]. For SBP in patients with cirrhosis and ascites, the reference standard was defined according to International Ascites Club consensus and incorporated in subsequent guidelines by the American Association for the Study of Liver Diseases (AASLD) and the European Association for the Study of the Liver (EASL), requiring an ascitic fluid polymorphonuclear leukocyte count ≥ 250 cells/mm^3^ irrespective of culture status [28,29]. To enable construction of 2 × 2 contingency tables, studies were required to report or facilitate derivation of true positives, false positives, false negatives, and true negatives for NGAL versus the reference standard.

We included prospective and retrospective cohort studies, cross-sectional surveys, and case–control designs published as full-text articles, as well as abstracts from scientific congresses published in indexed journals.

We pre-specified in PROSPERO the exclusion of case reports and very small case series (<10 participants) to mitigate extreme imprecision and sparse-data bias (e.g., zero-cell 2 × 2 tables) that can compromise the estimation and pooling of sensitivity and specificity and may impede convergence of hierarchical models. Accordingly, reports with <10 participants were not considered for quantitative synthesis, but would be described qualitatively if otherwise eligible. No language restrictions were imposed, and unpublished or gray-literature reports were considered when they provided sufficient diagnostic performance data. For the quantitative analysis, studies were excluded if they had not measured NGAL in peritoneal fluid, had not applied a peritonitis definition based on ISPD criteria, or had failed to report sufficient data to construct a 2 × 2 contingency table.

### 2.2. Search Strategy

We searched six bibliographic databases from inception to 31 December 2024 and performed an update search on 30 June 2025; the update identified no additional eligible studies. The databases were PubMed/MEDLINE, Embase, the Cochrane Library (CLIB), LILACS, Scopus, and the Web of Science Core Collection. We combined controlled vocabulary and free-text terms for neutrophil gelatinase-associated lipocalin (NGAL; “lipocalin-2”) and peritonitis (including “peritoneal dialysis,” “dialysis effluent,” “peritoneal fluid,” and “ascites”). Full database-specific strategies are provided in Table 1. We imposed no language or publication-status restrictions and also hand-searched reference lists and relevant reviews; we screened Google Scholar for conference abstracts/dissertations and queried ClinicalTrials.gov and the WHO ICTRP for ongoing or unpublished studies.

### 2.3. Study Selection

After duplicate removal, two reviewers (M.L.P-M and G.A.G.-F.) independently screened titles and abstracts against the eligibility criteria. Full texts of potentially relevant articles were subsequently assessed. Reference lists of included studies were also hand-searched. Disagreements were resolved by discussion or consultation with a third reviewer (J.D.G.-B).

### 2.4. Data Extraction and Quality Assessment of Studies

Two reviewers (M.L.P.-M. and V.A.C.-A.) independently extracted study-level characteristics (first author, publication year, study design, and geographic setting), patient demographics (mean or median age, sex distribution, and peritonitis subtype), index-test details (NGAL assay method and cut-off values), reference standard criteria, and 2 × 2 contingency data (true positives, false positives, false negatives, true negatives). Discrepancies were resolved by discussion and, when necessary, adjudicated by a third investigator (B.D.-V). Risk of bias and applicability concerns were evaluated using the QUADAS-2 tool, with each domain (patient selection, index test, reference standard, and flow and timing) rated as “low,” “high,” or “moderate” risk [26].

### 2.5. Statistical Analysis

For each study, we obtained a 2 × 2 table (true positives, false positives, false negatives, true negatives). When raw 2 × 2 data were unavailable, we reconstructed cell counts from reported accuracy metrics (sensitivity, specificity, and group totals) using the following algebraic reconstruction. Let *n_D_* and *n∼_D_* be the numbers of diseased and non-diseased participants, and let *Se* and *Sp* be the reported sensitivity and specificity (expressed as proportions). First compute the real-valued estimates*TP*\* = *Se* ⋅ *n_D_*, *TN*\* = *Sp* ⋅ *n∼_D_*

Select integer counts (*TP*,*TN*) by constrained integer rounding that minimizes the deviation from the reported accuracy:TP^,TN^=arg minTP,TN,∈ Z+[ TP/nD−Se2+(TN/ n∼D −Sp)2 ]
subject to 0 ≤ *TP* ≤ *n_D_* and 0 ≤ *TP* ≤ *n∼_D_*. In practice we evaluate the candidate set {⌊⋅⌋, round(⋅), ⌈⋅⌉} applied to *TP*\*, *TN*\* and choose the pair with the smallest squared error. Then set*FN* = *n_D_* − *TP*, *FP* = *n∼_D_* − *TN*.

We accepted a solution when ∣*TP*/*n_D_* − *Se*∣ ≤ 0.005 and ∣*TN*/*n∼_D_ − Sp*∣ ≤ 0.005 (±0.5 percentage points); otherwise, we keep the candidate with the minimal error. Column totals are preserved, non-negativity is enforced, and derived PPV/NPV (if reported) are cross-checked to agree within ±0.5 percentage points.

If a study reported multiple NGAL thresholds, we anchored the primary analysis to the study’s own choice: we preferentially extracted the prespecified clinical cut-off (e.g., manufacturer-recommended or stated in Methods); when no prespecification existed, we used the threshold employed in the article’s primary analysis (as highlighted in Results/tables). We did not re-optimize thresholds (e.g., maximize Youden) using our pooled data.

To control multiplicity and avoid unit-of-analysis errors, the primary bivariate random-effects model included one 2 × 2 table per study (one threshold per study). Any additional thresholds reported within a study were handled only in sensitivity/exploratory analyses (e.g., a multi-threshold HSROC specification) and were not pooled alongside the primary threshold. No confirmatory hypothesis testing was performed across multiple thresholds; our focus remained on estimation (SROC/HSROC surfaces and prediction regions).

All concentration units were harmonized to ng/mL. When articles reported different units (e.g., ng/dL), we applied exact conversions according to a prespecified data dictionary and documented these transformations in the study-level audit table.

The primary outcomes were pooled sensitivity and specificity and the area under the hierarchical summary ROC (HSROC) curve (including the partial AUC restricted to observed false-positive rates). Secondary summaries included likelihood ratios (LR+, LR−), the diagnostic odds ratio (DOR), and predictive values (PPV/NPV) calculated at pre-specified pre-test probabilities (15%, 20%, 25%, 30%, and 40%) via Fagan analysis. Beyond the primary bivariate random-effects synthesis, we derived pooled likelihood ratios (LR+, LR−) from the Reitsma model and used them to produce illustrative post-test probabilities via Fagan analysis across pre-specified pre-test probabilities (15%, 20%, 25%, 30%, 40%). In diagnostic test accuracy meta-analyses that include case–control designs, study-level “prevalence” is design-determined and not a valid estimator of real-world pre-test probability; therefore, we did not pool study-level prevalence. To aid clinical interpretation—without influencing primary estimates or inference—we report these Fagan-based updates as accessory summaries.

Diagnostic accuracy was synthesized using the bivariate random-effects model of Reitsma/Rutter–Gatsonis, which jointly models logit-transformed sensitivity and the logit-transformed false-positive rate (FPR = 1 − specificity) under a bivariate normal distribution, allowing between-study heterogeneity in both parameters and in their correlation. Models were fitted by restricted maximum likelihood (REML). We prespecified a random-effects framework because diagnostic accuracy was expected to vary across assays/platforms, thresholds, pre-analytical conditions (e.g., dwell time/handling), and patient spectra (PDAP vs. SBP). Fixed-effects assumptions were therefore implausible; the hierarchical bivariate model jointly accommodates variability in logit-sensitivity and logit-FPR and their correlation. Summary sensitivity and specificity on the probability scale were obtained by back-transforming the model’s logit-scale means. On the hierarchical summary receiver operating characteristic (HSROC) plot, we displayed the summary operating point with its 95% confidence and prediction regions, derived from the model’s variance–covariance components; in addition, we reported 95% prediction intervals for sensitivity and specificity to express expected variability in a new setting. For nested comparisons (e.g., adding subgroup covariates), models were re-estimated by maximum likelihood (ML) and compared using likelihood-ratio tests (LRTs). Between-study heterogeneity was described by the between-study variances (reported as logit-scale standard deviations for sensitivity and FPR) and their random correlation (rho, ρ). For completeness, we also report the corresponding variances (τ^2^) for each logit-scale standard deviation together with ρ, and we summarize dispersion on the probability scale using 95% prediction intervals and HSROC prediction regions.

Small-study effects were evaluated with Deeks’ funnel plot asymmetry test, regressing ln(DOR) on 1/√ESS (ESS = 4*n*_1_*n*_2_/*n*), after Haldane–Anscombe correction. We used weighted least-squares with weights proportional to ESS and considered *p* < 0.10 as suggestive of asymmetry, acknowledging the limited power with small *k*.

We examined prespecified subgroups by (i) clinical context (spontaneous bacterial peritonitis [SBP] vs. peritoneal dialysis-associated peritonitis [PDAP]), (ii) assay method (e.g., ELISA, automated immunoassay, rapid H-NGAL), (iii) testing platform (laboratory-based vs. point-of-care), and (iv) study size (<100 vs. ≥100 participants). For each factor, group differences in sensitivity and specificity were tested on the logit scale using random-effects meta-regressions with Knapp–Hartung small-sample adjustment; pooled Se/Sp are reported on the probability scale. In parallel, we fitted a bivariate Reitsma model including the subgroup as a covariate and compared it to the intercept-only model via a maximum-likelihood ratio test to assess the joint effect on logit-sensitivity and logit-FPR. As a secondary summary, between-group differences in diagnostic odds ratio were evaluated with random-effects meta-regression of log(DOR) using REML with Knapp–Hartung inference. Given the small number of studies in several strata, subgroup models were fitted one factor at a time (no multi-covariate adjustments), results were shown only when k ≥ 3 per stratum, and zero-cell studies used a 0.5 continuity correction (with 0.1 in sensitivity analyses).

The certainty of evidence for sensitivity, specificity, and HSROC AUC was appraised with GRADE for test accuracy, considering risk of bias (QUADAS-2), inconsistency (between-study heterogeneity), imprecision (width of 95% CIs), and publication bias (Deeks’ test). We did not downgrade for indirectness because all studies directly evaluated NGAL in peritoneal or ascitic fluid against accepted reference standards for peritonitis.

### 2.6. Data Visualization and Tabulation

We tabulated study-level characteristics (design, setting, patient population), assay details (method/platform, matrix, and positivity threshold), reference standards, and diagnostic 2 × 2 data with 95% CIs. Forest plots displayed individual and pooled estimates for sensitivity and specificity (and, as a secondary summary, DOR). HSROC plots showed the summary operating point with 95% confidence and prediction regions. Subgroups (SBP vs. PDAP) were visualized with stratified forest plots and separate HSROCs. Small-study effects were visualized with Deeks’ funnel plots (log-DOR vs. 1/√ESS).

### 2.7. Software and Statistical Packages

All analyses were performed in R (version 4.3.1) using RStudio. The mada package was used for the bivariate Reitsma model, HSROC summaries, Deeks’ test, and prediction intervals; metafor supported logit-scale random-effects meta-regressions with REML and Knapp–Hartung adjustments and provided influence diagnostics for log-DOR analyses. We used meta for complementary univariate summaries and forest plotting where appropriate, readxl for data import, dplyr for data management, and ggplot2 for figures. All code is available upon request.

## 3. Results

### 3.1. Study Selection

An electronic search of PubMed, Embase, Scopus, Web of Science, Cochrane Library, and LILACS retrieved 147 records. After removal of 101 duplicates, 46 unique titles and abstracts were screened, leading to the exclusion of 31 studies for ineligible population, intervention, or design. We obtained 15 articles for detailed assessment; of these, 1 was not accessible, and 1 lacked a valid reference standard or comparator. Ultimately, 13 full-text studies satisfied our inclusion criteria and were incorporated into the qualitative synthesis, with 10 studies providing sufficient data for the quantitative meta-analysis. A PRISMA flow diagram summarizing the selection process is shown in Figure 1.

### 3.2. Characteristics of Included Studies

Thirteen studies from between 2013 and 2024 met our inclusion criteria (Table 2 and Table 3), comprising two cross-sectional investigations, five case–control studies, two retrospective analyses (one with cross-sectional elements), one prospective observational study, two prospective cohort studies, and one case–cohort study with longitudinal follow-up. Collectively, these studies enrolled 1642 adults (sample sizes, 30–301; mean or median age approximately 45–65 years; sex distributions roughly balanced) across two clinical contexts: spontaneous bacterial peritonitis in cirrhotic ascites and peritoneal dialysis-associated peritonitis, from diverse geographic settings (Egypt, Italy, China, and the United States). All included studies enrolled at least 30 participants (range, 30–301); no study was excluded from the meta-analysis solely on the basis of the <10-participant rule.

Ascitic or peritoneal NGAL was measured with ELISA (n = 5), immunoturbidimetric/chemiluminescent automated assays (n = 4), rapid immunochromatographic tests (n = 1), and lateral-flow dipsticks (n = 2). Reported diagnostic thresholds ranged from ~85 to 300 ng/mL. All studies applied standardized reference standards (PMN ≥ 250/mm^3^ for SBP; ISPD criteria for PDAP).

Table 2 lists the 10 studies that contributed complete 2 × 2 data to the primary meta-analysis (one prespecified threshold per study). For two of these studies (Liu et al. [21] and Ahmed et al. [30], 2 × 2 counts were algebraically reconstructed from the reported sensitivity, specificity, and group totals, as detailed in Section 2. The remaining eight were directly extracted from the original articles. Table 3 presents the three studies without extractable 2 × 2 data (qualitative only; not included in the pooled analysis), while a comprehensive descriptive table of all 13 included studies is provided in Appendix A in the Appendix A.

### 3.3. Performance of Neutrophil Gelatinase-Associated Lipocalin for Detection of SBP and PDAP

Ten studies (573 cases; 833 controls) contributed to the meta-analysis. In the bivariate random-effects model, the pooled sensitivity was 0.95 (95% CI, 0.90–0.97) and specificity was 0.86 (0.70–0.94). To reflect between-study dispersion, 95% prediction intervals (PIs) indicated that a future study could observe a sensitivity around 0.75–0.98 and specificity around 0.23–0.99. Thus, while sensitivity appears consistently high across settings, specificity is highly variable, which limits generalizability and precludes stand-alone use. The HSROC AUC was ~0.95 (partial AUC restricted to observed FPRs 0.93) and the pooled DOR was ~90, but these summary metrics should be interpreted alongside the wide prediction regions. Forest plots for sensitivity, specificity, diagnostic odds ratio (DOR) and HSROC are shown in Figure 2a–d.

Between-study heterogeneity concentrated on specificity rather than sensitivity. In the overall bivariate model, the between-study SDs on the logit scale were 0.72 for sensitivity and 1.37 for the FPR, implying relatively stable sensitivity but markedly more variable specificity across settings. On the logit scale, between-study heterogeneity was τ^2^ = 0.61 (τ = 0.78) for sensitivity and τ^2^ = 2.19 (τ = 1.48) for specificity (univariate random-intercept models), corroborating that dispersion concentrates in specificity. Translating this to clinical terms using 95% PIs, a future study would be expected to observe sensitivity between 0.75 and 0.98 and specificity between 0.23 and 0.99.

For threshold and small-study effects, in the bivariate model, the random-effects correlation between sensitivity and the false-positive rate was negative (ρ ≈ −0.63), consistent with a threshold effect across studies. The scatter plot of logit-sensitivity versus logit-FPR also showed a negative Spearman correlation that did not reach statistical significance overall (ρ = −0.43; *p* = 0.218). Evidence of small-study effects was observed in the primary Deeks’ asymmetry test (*p* = 0.003). In a prespecified sensitivity analysis using an alternative continuity/weighting scheme, the signal attenuated (*p* ≈ 0.27), but given the small corpus, publication bias cannot be excluded, and we highlight this limitation throughout (Appendix A). Given the small number of studies and the influence of individual high-leverage studies, we interpret small-study effects as possible but not definitive.

Leave-one-out analyses of the pooled DOR supported robustness: estimates remained high across iterations (range, 58.43–124.71). Excluding Liu et al. [21] yielded the highest DOR (124.71; 95% CI, 32.51–478.45), whereas excluding Chen et al. [32] yielded the lowest (58.43; 95% CI, 14.96–228.18), without materially altering the conclusion (Figure 3).

For decision support, the pooled likelihood ratios were LR+ 6.96 and LR− 0.06. At a pre-test probability of 20%, a positive result increases the post-test probability to approximately 64%, whereas a negative result decreases it to about 1.5%. Because post-test probability depends on the underlying clinical pre-test probability, we provide illustrative updates using the pooled likelihood ratios. At a 10% pre-test probability, a positive NGAL result corresponds to a post-test probability of about 44%, while a negative result lowers it to ~0.7%. At a 20% pre-test probability, the corresponding values are ~64% and ~1.5% (as reported above). At a 40% pre-test probability, a positive result raises the post-test probability to ~82%, while a negative result reduces it to ~3.8%. However, these values are assay- and threshold-dependent and should be applied cautiously, considering the prediction intervals provided above.

### 3.4. Exploratory Subgroup Analysis by Peritonitis Type

Of the ten studies with quantitative data, six evaluated spontaneous bacterial peritonitis (n = 278 cases; n = 394 controls) and four evaluated peritoneal dialysis-associated peritonitis (n = 295 cases; n = 439 controls). Pooled accuracy was similar between groups (Figure 4), SBP: sensitivity 0.93 (95% CI, 0.86–0.97), specificity 0.85 (95% CI, 0.68–0.94), AUC 0.95. PDAP: sensitivity 0.96 (95% CI, 0.88–0.99), specificity 0.88 (95% CI, 0.50–0.98), AUC 0.94.

Formal subgroup tests found no evidence of a difference by peritonitis type. On the logit scale with REML and Knapp–Hartung adjustments, the difference in pooled sensitivity was *p* = 0.33 and in specificity *p* = 0.81. The DOR (PDAP/SBP) was 1.90 (95% CI, 0.07–52.42; *p* = 0.66), and a global LRT in the bivariate model ML detected no subgroup effect (*p* = 1.00).

Heterogeneity patterns were consistent but more extreme in PDAP. The between-study SDs were 0.31 (logit-sensitivity) and 1.94 (logit-FPR) for PDAP, yielding 95% PIs of Se 0.84–0.97 and Sp 0.07–>0.99 (the very wide Sp PI reflects the small k and boundary estimation). For SBP, SDs were 0.8 (logit-sensitivity) and 1.144 (logit-FPR), with PIs Se 0.68–0.99 and Sp 0.31–0.98. In univariate random-intercept models, between-study heterogeneity for sensitivity was τ^2^ = 0.53 (τ = 0.73) in SBP and τ^2^ = 0.47 (τ = 0.69) in PDAP. For specificity, heterogeneity was much larger—τ^2^ = 1.26 (τ = 1.12) in SBP and τ^2^ = 3.74 (τ = 1.94) in PDAP—confirming that dispersion concentrates in specificity, particularly for PDAP. Deeks’ test indicated small-study effects in SBP (*p* = 0.004) but not in PDAP (*p* = 0.16). Spearman’s correlations were negative but non-significant in both subgroups (SBP ρ = −0.66; *p* = 0.17; PDAP ρ = −0.80; *p* = 0.33).

Subgroup influence analyses mirrored the overall pattern (Figure 5). For PDAP, the pooled DOR was 133.17 (95% CI, 5.08–9509.62) with broader leave-one-out variation (30.74–315.59); excluding Chen et al. [32] gave the most conservative estimate (30.74; 95% CI, 5.12–184.54), while excluding Morisi et al. [38] produced the highest DOR (315.59; 95% CI, 0.28–353,806), with very wide CIs. For SBP, the pooled DOR was 77.32 (95% CI, 8.36–715.47) with tighter leave-one-out range (47.12–127.89); excluding Liu et al. [21] yielded the highest DOR (127.89; 95% CI, 36.84–443.95), and excluding Biomy et al. [31] the lowest (47.12; 95% CI, 4.87–455.78).

For clinical translation, subgroup LR estimates were PDAP LR+ 8.15/LR− 0.04 and SBP LR+ 6.16/LR− 0.08. These exploratory analyses (several strata with k ≤ 4) suggested broadly similar pooled accuracy between PDAP and SBP; however, prediction intervals for specificity remained wide, particularly in PDAP (e.g., ~0.07–>0.99), emphasizing threshold dispersion and limited power for between-group contrasts.

### 3.5. Exploratory Subgroup Analysis by NGAL Assay Method

We examined whether the assay methodology modifies diagnostic performance using a bivariate random-effects (Reitsma/HSROC) model stratified by assay type—ELISA (four studies), point-of-care/rapid tests (three studies), and automated immunoassays (three studies). These analyses were prespecified but exploratory and should be interpreted cautiously given the small number of studies per stratum and the presence of small-study effects at the meta-level.

In the bivariate model, ELISA yielded a pooled sensitivity of 0.94 (95% CI, 0.90–0.97) and specificity of 0.91 (0.84–0.95), corresponding to a DOR of 159.06 (55.33–457.25) and an HSROC AUC of 0.97. POC/rapid assays showed pooled sensitivity, 0.98 (0.94–0.99), and wide uncertainty in specificity, 0.90 (0.38–0.99); the resulting DOR was 242.91 (6.23–9475.85) and the AUC was 0.97. Automated immunoassays displayed lower overall accuracy, with sensitivity of 0.88 (0.75–0.94), specificity of 0.69 (0.49–0.84), DOR of 18.04 (2.27–143.36), and an AUC of 0.87.

Prediction intervals illustrated the extent of between-study dispersion. For sensitivity, the 95% PIs were narrow for ELISA (approximately 0.89–0.96) and POC/rapid (0.93–0.98), but wide for automated assays (0.58–0.97). For specificity, dispersion was modest for ELISA (about 0.79–0.95) and very large for both POC/rapid (0.03–0.99) and automated platforms (0.25–0.94), consistent with threshold variation across studies and explaining the broad HSROC prediction regions for these subgroups. Pooled accuracy and prediction intervals are shown in Figure 6.

In univariate random-intercept models stratified by assay, between-study heterogeneity on the logit scale was minimal for sensitivity with ELISA and POC/rapid (τ^2^ = 0.00; k = 4 and k = 3), but substantial for automated platforms (τ^2^ = 0.29, τ = 0.54; k = 3). For specificity, dispersion was modest for ELISA (τ^2^ = 0.12, τ = 0.35), and very large for POC/rapid (τ^2^ = 5.29, τ = 2.30) and automated assays (τ^2^ = 0.53, τ = 0.73). These results align with the very wide specificity prediction intervals and the HSROC prediction regions already shown for POC/rapid and automated platforms, indicating that assay choice materially contributes to between-study variability—particularly for specificity. Tests for subgroup differences were significant for both sensitivity (Q_between = 7.02, df = 2, *p* = 0.03) and specificity (Q_between = 7.45, df = 2, *p* = 0.02), reinforcing that platform differences are a meaningful source of heterogeneity, consistent with the HSROC results.

We collapsed platforms into two groups: laboratory-based assays (ELISA plus automated immunoassays) and point-of-care (POC) tests (Appendix A). The comparison included 10 studies overall: 7 laboratory-based studies (358 cases, 496 controls) and 3 POC studies (215 cases, 337 controls). Pooled estimates were obtained from a bivariate random-effects (Reitsma/HSROC) model; heterogeneity is described using prediction-region behavior and by leave-one-out (LOO) influence on the diagnostic odds ratio (DOR). Laboratory-based assays showed sensitivity 0.92 (95% CI 0.86–0.96) and specificity 0.84 (0.70–0.92), with DOR 67.54 (16.05–284.24) and AUC = 0.94. POC tests yielded a sensitivity of 0.97 (0.94–0.99) and specificity of 0.90 (0.38–0.99), with DOR of 242.91 (6.23–9475.85) and AUC = 0.97 (Appendix A).

Prediction regions from the HSROC displayed a narrow band for sensitivity in the POC subgroup and a broader band for specificity in both groups—extremely wide for POC, reflecting threshold dispersion across studies (Appendix A). This pattern mirrors the subgroup results by individual assay type and explains the large uncertainty around the POC DOR despite a high point estimate. A bivariate meta-regression likelihood ratio test confirmed that overall diagnostic accuracy differs between laboratory-based and POC tests (χ^2^ = 9.26, df = 2, *p* = <0.01). In keeping with the HSROC, this difference is driven primarily by the spread of specificity across POC studies and by the lower/heterogeneous specificity of automated assays within the laboratory group. (For reference, univariate subgroup tests on sensitivity and specificity alone did not reach conventional significance: χ^2^ ≈ 3.79, *p* ≈ 0.052 for sensitivity; χ^2^ ≈ 0.14, *p* ≈ 0.71 for specificity.)

As detailed in Appendix A in the Appendix A, POC results are highly sensitive to individual studies: omitting Chen et al. [32] reduced the POC DOR to 31.6, whereas omitting Morisi et al. [38], increased it to 1253.0 and omitting Virzi et al. [35] raised it to 351.8. Within the laboratory group, influence is concentrated in the automated-assay subset: excluding Liu et al. [21], increased the automated DOR to 46.1, while excluding Lippi et al. [20], and Martino et al. [22] lowered it to 10.4 and 12.3, respectively. ELISA-based estimates were comparatively robust, with smaller LOO shifts. Together, these findings reinforce that assay choice and study-level thresholds materially affect observed performance.

Deeks’ funnel asymmetry tests showed no evidence of small-study effects for POC (*p* = 0.727) and a borderline signal for the laboratory group (*p* = 0.062), advising cautious interpretation of between-group contrasts.

### 3.6. Exploratory Subgroup Analysis by Sample Size

To assess whether study size affects diagnostic performance, we stratified studies by total sample size using a threshold of 100 participants (Figure 7). Four studies with <100 participants contributed 135 peritonitis cases and 132 controls, and six studies with ≥100 participants contributed 438 cases and 701 controls. The pooled accuracy in the bivariate model for small studies (<100) showed sensitivity 0.957 (95% CI 0.90–0.98) and specificity 0.92 (0.86–0.96), yielding DOR 245.81 (82.78–729.96) and AUC 0.97. Large studies (≥100) showed sensitivity 0.95 (0.86–0.97) and specificity 0.81 (0.50–0.94), with DOR 50.22 (7.80–323.55) and AUC 0.94 (Appendix A).

The HSROC display shows a narrower prediction region for sensitivity in small studies and a markedly broader region for specificity—especially among larger studies—indicating substantial threshold dispersion for specificity (Figure 7d). These patterns overlap widely across size strata, consistent with the non-significant LRT.

LOO analyses indicated moderate influence within the <100 group (baseline DOR = 245.81): omitting Biomy et al. [31] lowered the DOR to 174.1, whereas omitting Ahmed et al. [30], Hassan et al. [19], or Virzi et al. [37] raised it to 269.9, 353.5, and 260.6, respectively. In the ≥100 group (baseline DOR = 50.22), omitting Chen et al. [32] reduced the DOR to 22.7; excluding Liu et al. [21] increased it to 84.5; other omissions had smaller effects (e.g., Martino et al. [36] 57.0, Morisi et al. [38], 67.6, Lippi et al. [36], 50.0, Khalil et al. [33] 51.2). Overall, large-study results are more sensitive to single influential studies, largely through specificity.

Deeks’ funnel asymmetry tests did not suggest small-study effects within either stratum (*p* = 0.59 for <100; *p* = 0.46 for ≥100). Hence, although smaller studies show higher pooled DOR and AUC, these features are not explained by publication bias in the Deeks sense.

The wider HSROC prediction region for specificity among larger studies—and the pronounced LOO shifts when omitting certain large studies—suggest that between-study threshold differences and assay mix (e.g., inclusion of low-specificity automated platforms) likely contribute to the observed patterns.

Complementing the HSROC prediction regions and LOO diagnostics, univariate random-intercept models on the logit scale showed that between-study heterogeneity for sensitivity was essentially absent among smaller studies (<100 participants; τ^2^ = 0.00, τ = 0.00) but appreciable among larger studies (≥100; τ^2^ = 0.73, τ = 0.86). For specificity, dispersion was again minimal in smaller studies (τ^2^ < 0.01, τ < 0.01) and strikingly greater in larger studies (τ^2^ = 2.78, τ = 1.67). For the diagnostic odds ratio, heterogeneity was low in smaller studies (τ^2^ < 0.01, τ ≈ 0.01) and high in larger studies (τ^2^ = 1.73, τ = 1.32). Consistent with these patterns, the between-group test was significant for DOR (Q_between = 7.78, *p* = < 0.01), whereas sensitivity (Q_between = 0.80, *p* = 0.37) and specificity (Q_between = 1.96, *p* = 0.16) did not differ significantly between size strata. In the bivariate Reitsma framework; however, a meta-regression did not detect a joint effect of study size on accuracy (χ^2^ = 1.76, df = 2, *p* = 0.42), and the HSROC prediction regions overlapped widely—indicating that the observed contrasts are driven primarily by specificity dispersion and the assay mix within the larger-study stratum rather than a uniform size effect per se.

### 3.7. Risk of Bias

During the screening process, four independent authors (J.C.D.A.-J, J.B.-R, M.L.P.-M. and J.D.G.-B) demonstrated a high degree of agreement, as supported by a Cohen’s kappa coefficient of 0.95. The methodological quality of the included studies was assessed using the QUADAS-2 tool with a domain-level bar chart (Figure 8). Overall, the risk of bias was judged to be low to moderate across the included studies. The primary concerns identified were related to the ‘Patient Selection’ domain, particularly in studies employing a case–control design, which may introduce a risk of selection bias.

Deeks’ funnel plot for all studies showed evidence of small-study effects (slope test *p* = 0.003; Figure 9a). In subgroup analyses by assay method, no asymmetry was detected for ELISA (k = 4, *p* = 0.56), automated immunoassays (k = 3, *p* = 0.67), or point-of-care (POC) tests (k = 3, *p* = 0.73) (Figure 9b). Grouping by platform indicated borderline asymmetry among laboratory-based assays (k = 7, *p* = 0.062), whereas POC tests again showed no signal (k = 3, *p* = 0.73) (Figure 9c). Stratifying by study size revealed no asymmetry within either stratum (<100 participants: k = 4, *p* = 0.585; ≥100 participants: k = 6, *p* = 0.46) (Figure 9d). Taken together, the significant overall result appears to be driven by the set of laboratory-based studies; all subgroup findings should be interpreted cautiously given the small number of studies per stratum (k ≤ 4) and the conventional *p* < 0.10 threshold used for Deeks’ test.

### 3.8. Certainty of the Evidence

The GRADE–DTA evaluation, conducted independently by four authors (J.A.G.-F., A.M.L.-Y., J.C.D.A.-J., and J.B.-R.), of thirteen diagnostic performance studies assessing NGAL in peritoneal effluent and ascitic fluid is summarized in Table 4. The risk of bias was generally moderate across studies, with inconsistency, indirectness, and publication bias judged as not serious. However, serious imprecision—due largely to wide or unreported confidence intervals—resulted in moderate overall certainty for most studies. Only Chen et al. [32], achieved high certainty on the basis of narrow estimates and rigorous design, while Lippi et al. [20] maintained moderate certainty despite the absence of serious imprecision, likely reflecting their moderate risk of bias assessment.

## 4. Discussion

In this systematic review and meta-analysis, peritoneal/ascitic neutrophil gelatinase-associated lipocalin (NGAL) showed high pooled sensitivity with context-dependent specificity for peritonitis. In the bivariate random-effects model, pooled sensitivity and specificity were 0.95 and 0.86, respectively; importantly, 95% prediction intervals indicated that a future study could observe sensitivity around 0.75–0.98 and specificity around 0.23–0.99, underscoring substantial threshold and setting dispersion. HSROC AUC (~0.95), DOR (~90), and likelihood ratios (LR+ ~7.0, LR− 0.06) should therefore be interpreted alongside these wide prediction regions rather than in isolation.

Diagnostic performance appeared broadly similar between peritonitis types. While formal subgroup tests did not show differences, prediction intervals—particularly for specificity—remained wide (e.g., PDAP specificity PI ~0.07–>0.99), reflecting limited power (k ≤ 4 in strata) and threshold variability. These findings support NGAL as an adjunct across PDAP and SBP, not as a stand-alone rule-in or rule-out test.

A platform-level contrast was detected, largely driven by specificity dispersion across studies and by variability in automated assays. Given these patterns and the limited numbers within subgroups, assay-specific, two-threshold strategies (low threshold for high-sensitivity rule-out; higher threshold for rule-in) are hypothesis-generating at this moment and require local verification before implementation.

The risk of bias was generally low to moderate according to QUADAS-2, with anticipated concerns for spectrum bias in case–control designs. At the meta-level, Deeks’ funnel plot showed evidence of small-study effects (*p* = 0.003); although a prespecified sensitivity analysis attenuated this (*p* ≈ 0.27), publication/small-study bias cannot be excluded. A sample-size meta-regression did not detect a significant size effect on joint accuracy, yet the small corpus and influential studies counsel caution. The overall certainty according to GRADE-DTA was moderate due to imprecision and design limitations.

Regarding clinical use, current guidelines for PDAP and SBP rely on clinical features, PMN thresholds, and cultures, which can be delayed or affected by pre-analytics (e.g., dwell time) [27,28,29]. Our synthesis supports NGAL as a rapid adjunct within these pathways. The attractive rule-out profile (low LR−) must nevertheless be interpreted with the wide specificity PIs in mind, and local validation of thresholds, pre-analytic standardization (e.g., ≥2-h dwell for PD effluent), and quality control/training—especially for POC assays—are prerequisites. POC formats (cassette/strip/pen) can be read at the bedside in ~10–15 min, fitting triage workflows when PMN counts and cultures are pending. Published evaluations show good agreement with laboratory NGAL and with effluent WCC, and acceptable inter-operator reproducibility (e.g., correlation with lab NGAL and WCC, and κ for operator agreement); however, until prospective impact and cost-effectiveness evaluations are available, recommendations should remain cautious.

Key limitations include substantial heterogeneity in specificity, small sample sizes in several strata, the need for 2 × 2 table reconstruction in two studies, potential spectrum bias due to case–control sampling, and limited peer review in some data sources, all of which further temper the strength of the conclusions [37,38]. Our a priori exclusion of <10-participant series was intended to reduce sparse-data artifacts in hierarchical modeling. Notably, the decision had no practical impact because all eligible studies had ≥30 participants. Furthermore, our prespecified exploratory sample-size analysis found no significant effect of study size on the joint sensitivity/specificity summary, although the small number of studies per stratum limits power.

A further limitation is false-positive elevation in non-infectious inflammation. NGAL rises with sterile peritoneal inflammation. NGAL is produced by activated neutrophils and injured epithelial cells and is overexpressed in several cancers; elevations have been documented in ovarian cancer tissues and patient sera [39]. Malignant ascites are enriched by tumor- and inflammation-related proteins and vesicles, which may elevate NGAL concentrations even in the absence of infection [40]. Similarly, acute pancreatitis—an archetypal sterile inflammatory condition—demonstrates early increases in serum and urine NGAL levels, suggesting that pancreatitis-associated ascites could also exhibit elevated baseline NGAL concentrations [41]. Beyond assay and threshold variability, baseline inflammatory milieu differs by population: patients with end-stage kidney disease on peritoneal dialysis often exhibit chronic peritoneal/systemic microinflammation, whereas decompensated cirrhosis entails immune dysfunction with sustained systemic inflammation [42,43,44]. These background differences plausibly raise baseline NGAL in the absence of infection, contributing to the wide prediction intervals—especially for specificity—and potential spectrum bias when pooling PDAP and SBP. Accordingly, population-specific, locally validated thresholds are preferable to a universal cut-off. Therefore, SBP/PDAP cut-offs should not be extrapolated to non-cirrhotic etiologies without validation; a two-threshold or composite algorithm (NGAL + PMN + routine chemistries) with explicit gray-zone reflex testing is advisable.

Future research should focus on (i) prospective multicenter validations using prespecified, assay-specific two thresholds with blinded adjudication and real-world case mix; (ii) pre-analytic and analytical standardization (dwell time, handling, LoD/LoQ, imprecision, matrix effects) to enable comparability; (iii) impact/implementation and economic studies (time-to-antibiotics, catheter outcomes, LOS, readmissions, mortality; budget impact, price-thresholds for POC); and (iv) etiology-specific evaluations of sterile inflammatory ascites (malignancy, pancreatitis) to define incremental value beyond PMN and routine chemistries. Future multicenter studies with individual patient data would enable a full decision-curve analysis (DCA) meta-analysis, providing a robust framework to quantify net clinical benefit and directly inform guideline-based diagnostic pathways.

Integration with guideline-based criteria remains essential. If locally validated, bedside NGAL testing may help accelerate triage while confirmatory diagnostics (PMN count and cultures) are pending, and combinatorial approaches with other biomarkers warrant further evaluation. Given the wide prediction intervals—particularly for specificity—and the observed platform- and threshold-dependent behavior, post-test interpretation should be anchored to the local pre-test probability. Centers should estimate this using recent program data (e.g., the proportion of “suspected peritonitis” encounters that fulfilled ISPD, AASLD, or EASL reference standards). Once pre-analytical and analytical procedures, as well as assay-specific thresholds, are standardized and locally verified, the pooled likelihood ratios can be applied to derive site-specific post-test probabilities. Any prognostic applications (e.g., incorporating NGAL into MELD-Na or infection-risk scores) should proceed only after diagnostic pathways are standardized, clinically validated, and embedded within guideline-concordant algorithms.

## 5. Conclusions

NGAL measured in peritoneal or ascitic fluid is a promising adjunct, showing consistently high sensitivity but context-dependent specificity; however, the wide between-study variability—linked to threshold choice, assay/platform differences, pre-analytic conditions, and population mix—together with spectrum bias in case–control designs and signals of small-study effects, warrants cautious interpretation and precludes stand-alone rule-out or rule-in use. Clinical application should therefore rely on assay-specific, locally verified thresholds embedded within guideline-concordant pathways, with standardized dwell time/handling, robust quality control (especially for point-of-care formats), and explicit gray-zone or composite algorithms that anticipate false positives in sterile peritoneal inflammation. Priorities for next steps include prospective multicenter validations under standardized analytical and pre-analytical conditions and impact/economic evaluations to establish where NGAL meaningfully accelerates care and improves outcomes without compromising antimicrobial stewardship.

## Figures and Tables

**Figure 1 medsci-13-00175-f001:**
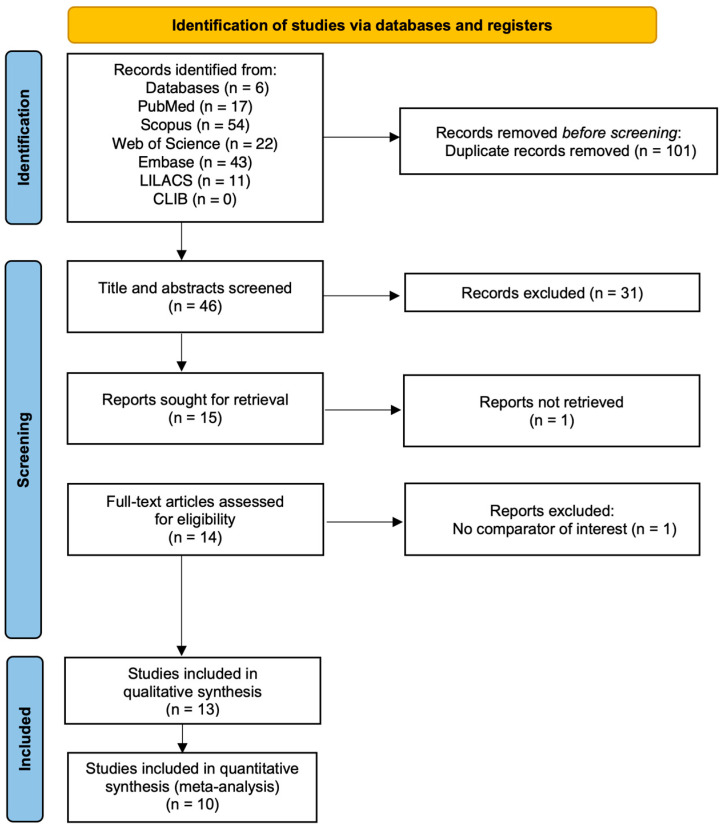
PRISMA flow diagram.

**Figure 2 medsci-13-00175-f002:**
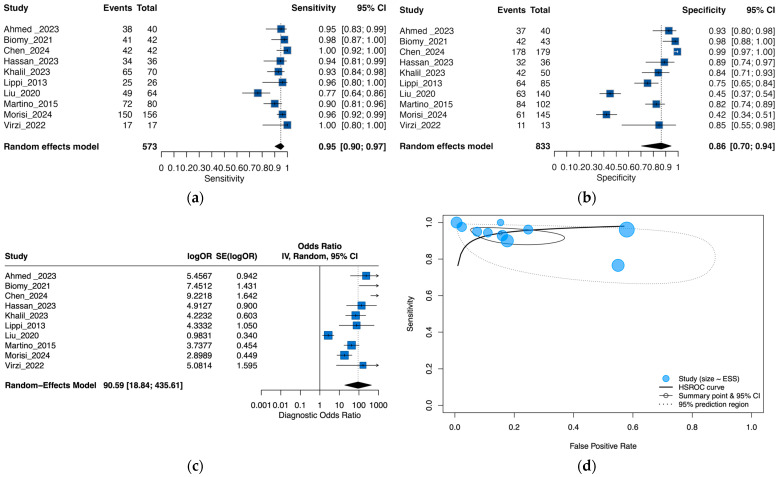
Forest plots of diagnostic performance metrics for NGAL in peritonitis diagnosis: (**a**) sensitivity; (**b**) specificity; (**c**) diagnostic odds ratio; (**d**) summary receiver operating characteristic curve for NGAL diagnostic performance [19,20,21,22,30,31,32,33,34].

**Figure 3 medsci-13-00175-f003:**
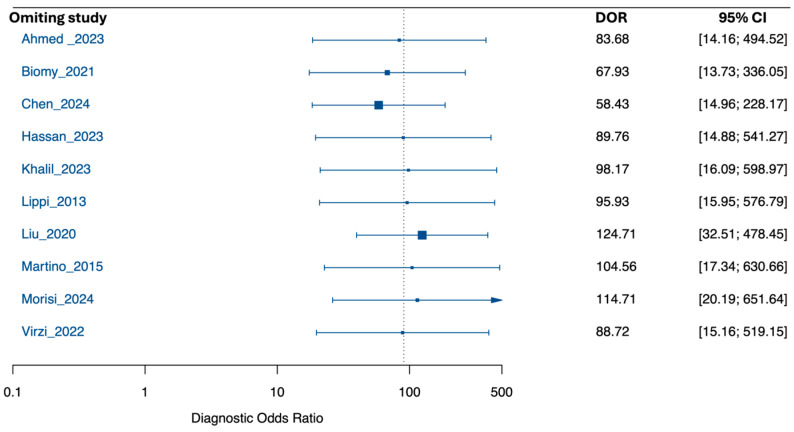
Leave-One-Out sensitivity analysis of the pooled diagnostic odds ratio for NGAL in peritonitis diagnosis [19,20,21,22,30,31,32,33,34].

**Figure 4 medsci-13-00175-f004:**
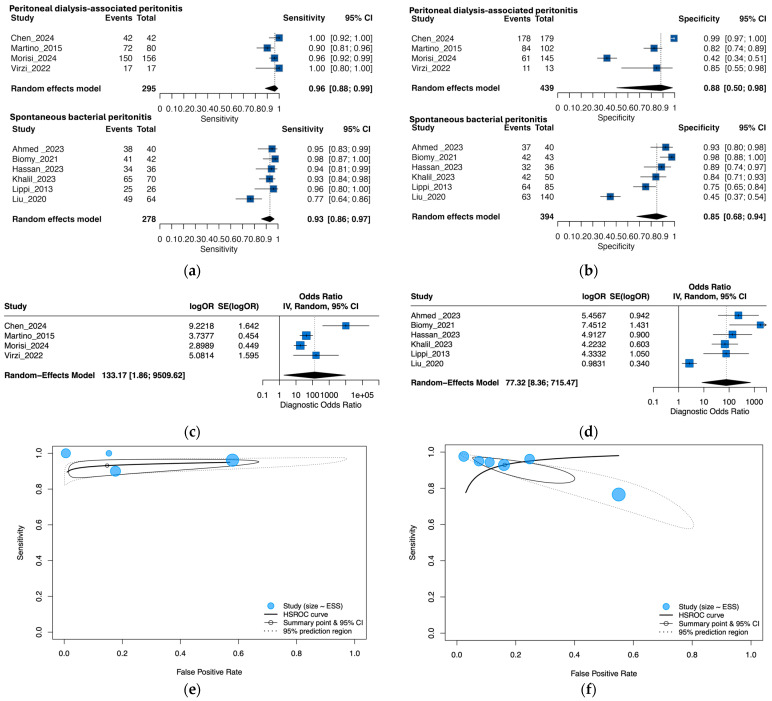
Subgroup analysis of NGAL diagnostic performance by peritonitis type: (**a**) forest plot of sensitivity by peritonitis type; (**b**) forest plot of specificity by peritonitis type; (**c**) PDAP diagnostic odds ratios; (**d**) SBP diagnostic odds ratios; (**e**) summary of receiver operating characteristic curves in PDAP; (**f**) summary of receiver operating characteristic curves in SBP [19,20,21,22,30,31,32,33,34,35].

**Figure 5 medsci-13-00175-f005:**
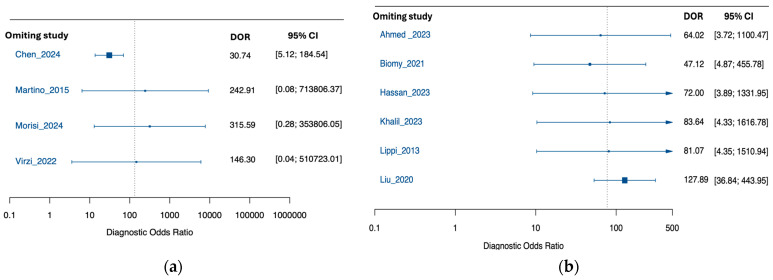
Leave-One-Out sensitivity analysis of the pooled diagnostic odds ratio for NGAL in peritonitis diagnosis: (**a**) peritoneal dialysis-associated peritonitis; (**b**) spontaneous bacterial peritonitis [19,20,21,22,30,31,32,33,34,35].

**Figure 6 medsci-13-00175-f006:**
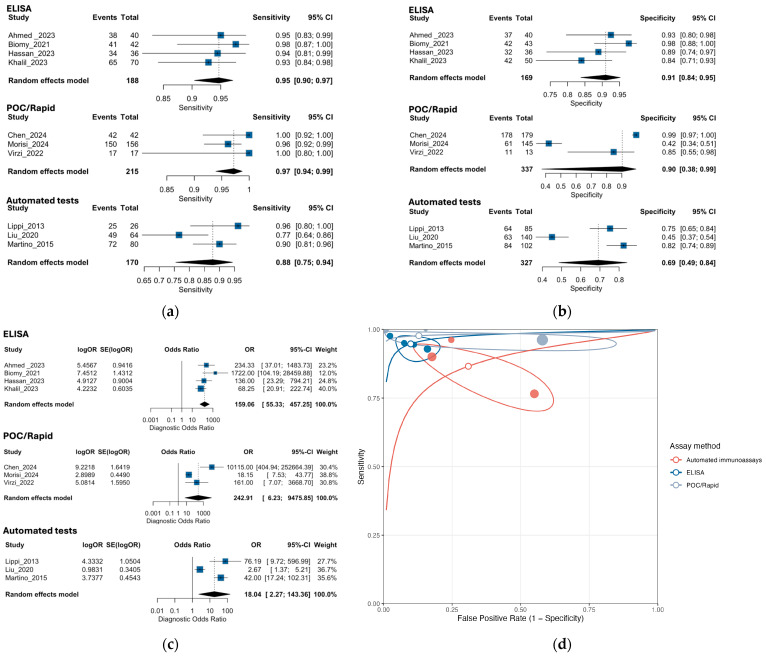
Subgroup analysis of NGAL diagnostic performance by NGAL test type: (**a**) forest plot of sensitivity by NGAL test type; (**b**) forest plot of specificity by NGAL test type; (**c**) forest plot of diagnostic odds ratio by NGAL test type; (**d**) summary receiver operating characteristic (SROC) curves by NGAL test type [19,20,21,22,30,31,32,33,34,35].

**Figure 7 medsci-13-00175-f007:**
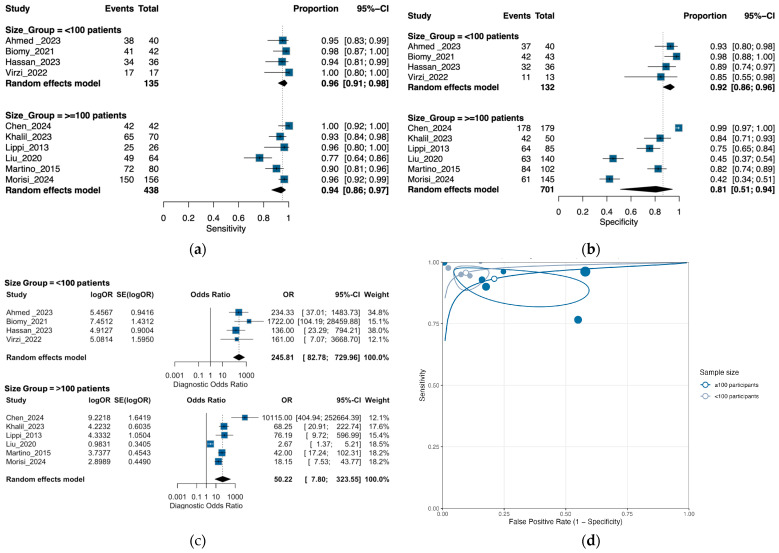
Subgroup analysis of NGAL diagnostic performance by NGAL study sample size: (**a**) forest plot of sensitivity by NGAL study sample size; (**b**) forest plot of specificity by NGAL study sample size; (**c**) forest plot of diagnostic odds ratio by NGAL study sample size; (**d**) summary receiver operating characteristic curves by study sample size [19,20,21,22,30,31,32,33,34,35].

**Figure 8 medsci-13-00175-f008:**
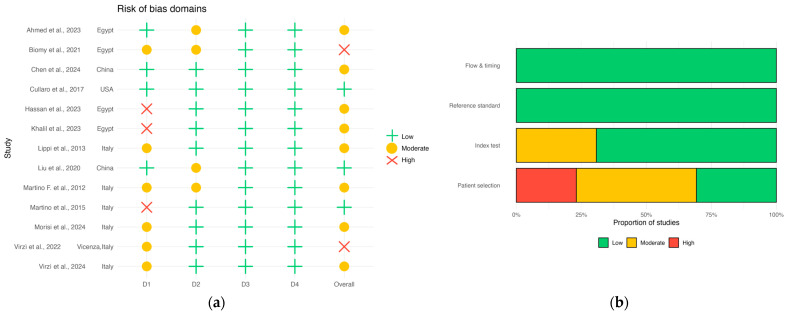
QUADAS-2 risk-of-bias assessment across included studies. (**a**) Traffic-light plot showing study-level judgments for the four QUADAS-2 domains: D1 patient selection, D2 index test, D3 reference standard, and D4 flow and timing, plus the overall judgment. (**b**) Domain-level stacked bar chart summarizing the proportion of studies rated low, moderate, or high for each domain [13,19,20,21,22,30,31,32,33,34,35,36,37].

**Figure 9 medsci-13-00175-f009:**
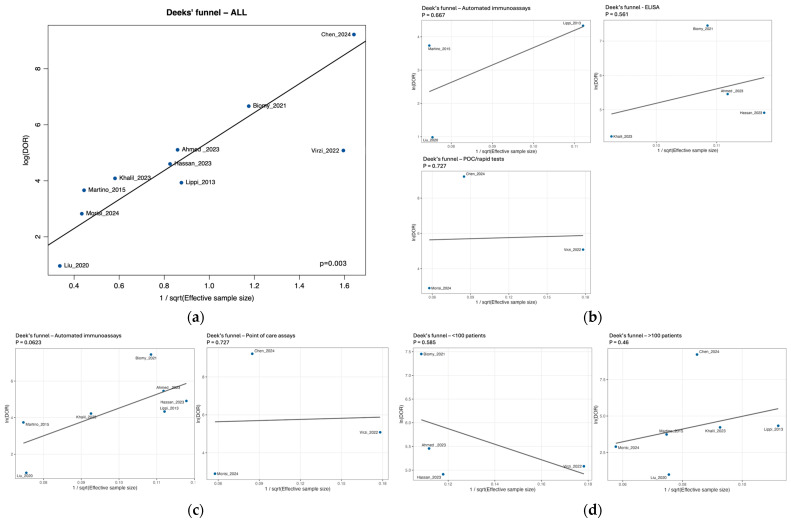
Deeks’ funnel plots for ascitic/peritoneal NGAL. (**a**) All studies (slope test *p* = 0.003). (**b**) By assay method: automated immunoassays (*p* = 0.67), ELISA (*p* = 0.56), point-of-care tests (*p* = 0.73). (**c**) By platform: laboratory-based assays (*p* = 0.06) vs. POC (*p* = 0.73). (**d**) By study size: <100 (*p* = 0.59) vs. ≥100 participants (*p* = 0.46). Each panel regresses ln(DOR) on 1/√ESS; *p* values correspond to Deeks’ asymmetry test (lower *p* suggests small-study effects).

**Table 1 medsci-13-00175-t001:** Database-specific search strategies for NGAL and peritonitis diagnostics.

Database	Search Strategy
PubMed/MEDLINE	(“neutrophil gelatinase-associated lipocalin”[Title/Abstract] OR “NGAL”[Title/Abstract] OR “lipocalin-2”[Title/Abstract]) AND (“peritoneal dialysis”[Title/Abstract] OR “dialysis effluent”[Title/Abstract] OR “peritoneal fluid”[Title/Abstract] OR “ascites”[Title/Abstract]) AND (“peritonitis”[Title/Abstract] OR “spontaneous bacterial peritonitis”[Title/Abstract] OR “secondary bacterial peritonitis”[Title/Abstract] OR “infection”[Title/Abstract])
Embase	(‘neutrophil gelatinase associated lipocalin’:ab,ti OR ‘NGAL’:ab,ti OR ‘lipocalin 2’:ab,ti) AND (‘peritoneal dialysis’:ab,ti OR ‘dialysis effluent’:ab,ti OR ‘peritoneal fluid’:ab,ti OR ‘ascites’:ab,ti) AND (‘peritonitis’:ab,ti OR ‘spontaneous bacterial peritonitis’:ab,ti OR ‘secondary bacterial peritonitis’:ab,ti OR ‘infection’:ab,ti)
Cochrane Library	(“neutrophil gelatinase-associated lipocalin” OR NGAL OR “lipocalin-2”) AND (“peritoneal dialysis” OR “dialysis effluent” OR “peritoneal fluid” OR “ascites”) AND (peritonitis OR “spontaneous bacterial peritonitis” OR “secondary bacterial peritonitis” OR infection)
LILACS	(“neutrophil gelatinase-associated lipocalin” OR NGAL OR “lipocalin-2”) AND (“diálisis peritoneal” OR “efluente peritoneal” OR “líquido peritoneal” OR ascitis) AND (peritonitis OR “peritonitis bacteriana espontánea” OR “peritonitis bacteriana secundaria” OR infección)
Scopus	TITLE-ABS-KEY(“neutrophil gelatinase-associated lipocalin” OR NGAL OR “lipocalin-2”) AND TITLE-ABS-KEY(“peritoneal dialysis” OR “dialysis effluent” OR “peritoneal fluid” OR ascites) AND TITLE-ABS-KEY(peritonitis OR “spontaneous bacterial peritonitis” OR “secondary bacterial peritonitis” OR infection)
WoS ^1^	TS = (“neutrophil gelatinase-associated lipocalin” OR NGAL OR “lipocalin-2”) AND TS = (“peritoneal dialysis” OR “dialysis effluent” OR “peritoneal fluid” OR ascites) AND TS = (peritonitis OR “spontaneous bacterial peritonitis” OR “secondary bacterial peritonitis” OR infection)

^1^ Wos. Web of Science.

**Table 2 medsci-13-00175-t002:** Studies contributing complete 2 × 2 data to the primary meta-analysis (one threshold per study).

Study (Year, Country; Context)	Design	NGAL Method	Threshold Used in Primary Study	TP	FP	TN	FN	Diagnostic Performance Metrics Reported	QUADAS-2 (Overall)
Ahmed et al., 2023 [30] (Egypt); SBP †	Case–control	ELISA (SunRed); laboratory	297.80 ng/mL	38	3	37	2	-Sensitivity: 95.6% -Specificity: 92.5% -Positive Predictive Value: 95% -Negative Predictive Value: 95% -AUC: 0.845 -Accuracy: 95%	Moderate
Biomy et al., 2021 [31], (Egypt); SBP	Cross-sectional	ELISA (Bioassay Science Lab E1719Hu); laboratory	100.80 ng/mL	41	1	42	1	-Sensitivity: 97.62% -Specificity: 97.67% -Positive predictive value: 97.62% -Negative predictive value: 97.67% -AUC: 0.974	High
Chen et al., 2024 [32], (China); PDAP	Prospective diagnostic	Rapid immunochromatographic POC	Positive/negative per kit ‡	42	1	178	0	-Sensitivity: 100% (95% CI 91.62–100%) -Specificity: 99.44% (95% CI 96.90–99.90%) -Accuracy: 99.55% (95% CI 97.48–99.92%) -Positive Predictive Value: 97.67% (95% CI 87.94–99.59%) -Negative Predictive Value: 100% (95% CI 97.89–100%) -Kappa value: 0.985 (95% CI 0.956–1.000)	Moderate
Hassan et al., 2023 [19], (Egypt); SBP	Case–control	ELISA (SunRed); laboratory	230.05 ng/mL	34	4	32	2	-Sensitivity: 94.4% -Specificity: 88.9% -Positive Predictive Value: 89.5% -Negative Predictive Value: 94.1% -AUC: 0.989- Accuracy: 91.7%	Moderate
Khalil et al., 2023 [33], (Egypt); SBP	Case–control	ELISA (DRG GmbH); laboratory	110.72 ng/mL	65	8	42	5	-Sensitivity: 92.7% (CI: [0.82, 0.98]) -Specificity: 84% (CI: [0.75, 0.91]) -Positive Predictive Value: 0.77 (CI: [0.65, 0.86]) -Negative Predictive Value: 0.95 (CI: [0.88, 0.98]) -Area under the ROC curve (AUC): 0.899 (CI: [0.848, 0.951])	Moderate
Lippi et al., 2013 [20], (Italy); SBP	Cross-sectional	Immunoturbidimetric (BioPorto NGAL Test™); laboratory	120.0 ng/mL	25	21	64	1	-Sensitivity: 96% (95% CI: 80–100%) -Specificity: 75% (95% CI: 65–84%) -Positive Predictive Value: Not reported -Negative Predictive Value: Not reported -AUC: 0.89 (95% CI: 0.82–0.95)	Moderate
Liu et al., 2020 [21], (China); SBP †	Prospective cohort	Immunoturbidimetric (BSBE); laboratory	108.95 ng/mL	49	77	63	15	-Sensitivity: 76.9% (for mortality prediction) -Specificity: 45.1% (for mortality prediction) -Positive Predictive Value: Not reported -Negative Predictive Value: Not reported	Low
Martino et al., 2015 [22], (Italy); PDAP	Case–control	CMIA (Abbott Architect); laboratory	85.00 ng/mL	72	18	84	8	-Sensitivity: Not reported -Specificity: Not reported -Positive Predictive Value: Not reported -Negative Predictive Value: Not reported -AUC peritoneal NGAL 0.99	Low
Morisi et al., 2024 [34], (Italy); PDAP	Retrospective diagnostic	POC dipstick (NGALds)	100 ng/mL	150	84	61	6	-Sensitivity: 96% -Specificity: Not reported -Positive predictive value: 0.64 -Negative predictive value: 0.87 -Area under the ROC curve (AUC): 0.82	Moderate
Virzì et al., 2022 [35], (Italy); PDAP	Case–control	POC dipstick (BioPorto NGALds) + lab comparator	300 ng/mL	17	2	11	0	Spearman’s ρ between NGALds and lab NGAL = 0.88 (*p* < 0.01), Spearman’s ρ between NGALds and effluent WCC = 0.82 (*p* < 0.01), Inter-operator reproducibility: ρ = 0.847 (*p* < 0.001), κ = 0.786 (*p* < 0.001).	High

“SBP” = spontaneous bacterial peritonitis; “PDAP” = peritoneal dialysis-associated peritonitis. All quantitative thresholds are standardized to ng/mL; qualitative POC assays are denoted as positive/negative per the manufacturer’s read. 2 × 2 sources: Liu et al. [21] and Ahmed et al. [30] were reconstructed (†) per the algebraic procedure specified in Section 2; the other eight were directly extracted from the reports.

**Table 3 medsci-13-00175-t003:** Studies without extractable 2 × 2 data (qualitative only; not pooled) (n = 3).

Study	Design/Context	NGAL Method/Platform	Threshold as Reported	Diagnostic Performance Metrics Reported	QUADAS-2 (Overall)
Cullaro et al., 2017 [13], (USA)	Prospective cohort; SBP	ELISA (laboratory; AntibodyShop)	Diagnostic cut-off for SBP not prespecified (AUC reported)	-For SBP diagnosis: -c-statistic (AUC): 0.68	Low
Martino et al., 2012 [36], (Italy)	Case–control; PDAP	Chemiluminescent immunoassay (Abbott Architect)	Threshold not specified (ROC performed)	-AUC peritoneal NGAL 0.99	Moderate
Virzì et al., 2024 [37], (Italy)	Retrospective; PDAP	Laboratory turbidimetric (NGALlab) and rapid dipstick (NGALds)	Lab: >100 µg/L; Dipstick categories 25–600 µg/L	-Spearman’s Rs = 0.876 (*p* = 10^−96^) -Sensitivity: 96% (150/156)	Moderate

**Table 4 medsci-13-00175-t004:** Summary of GRADE–DTA domains and overall certainty for studies on NGAL as a diagnostic biomarker in peritoneal effluent and ascitic fluid.

Study (Year)	Risk of Bias	Inconsistency	Indirectness	Imprecision	Publication Bias	Overall Certainty
Ahmed et al. [30], 2023	Moderate	Not serious	Not serious	Serious	Undetected	Moderate
Biomy et al. [31], 2021	Moderate	Not serious	Not serious	Serious	Undetected	Moderate
Chen et al. [32], 2024	Not serious	Not serious	Not serious	Not serious	Undetected	High
Cullaro et al. [13], 2017	Moderate	Not serious	Not serious	Serious	Undetected	Moderate
Hassan et al. [19], 2023	Moderate	Not serious	Not serious	Serious	Undetected	Moderate
Khalil et al. [33], 2023	Moderate	Not serious	Not serious	Serious	Undetected	Moderate
Lippi et al. [20], 2013	Moderate	Not serious	Not serious	Not serious	Undetected	Moderate
Liu et al. [21], 2020	Moderate	Not serious	Not serious	Serious	Undetected	Moderate
Martino et al. [36], 2012	Moderate	Not serious	Not serious	Serious	Undetected	Moderate
Martino et al. [22], 2015	Moderate	Not serious	Not serious	Serious	Undetected	Moderate
Morisi et al. [34], 2024	Moderate	Not serious	Not serious	Serious	Undetected	Moderate
Virzì et al. [35], 2022	Moderate	Not serious	Not serious	Serious	Undetected	Moderate
Virzì et al. [37], 2024	Moderate	Not serious	Not serious	Serious	Undetected	Moderate

## Data Availability

The original contributions presented in this study are included in the article/Appendix A. Further inquiries can be directed to the corresponding author(s).

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
