# Peer review of "Diagnostic Accuracy of Neutrophil Gelatinase-Associated Lipocalin in Peritoneal Effluent and Ascitic Fluid for Early Detection of Peritonitis: A Systematic Review and Meta-Analysis"

_medsci, 2025, doi:10.3390/medsci13030175_

Round 1

Reviewer 1 Report

Comments and Suggestions for Authors

I read with attention Manuel Luis Prieto-Magallanes et al's manuscript, concerning interest of NGAL concentration in infection diagnosis in case of SBP or dialysis-associated peritonitis. They wrote an excellent and complete paper with a clear description of methods.

I have only one comment concerning discussion part: NGAL could be interesting to be studied in case of non-ascitic ascites, notably in case of tumour for infection diagnosis and in case of pancreatitis associated ascites, because of associated inflammation, that can increase WBC.

Author Response

We are deeply grateful to the editor and reviewers for their careful evaluation of our manuscript. The detailed and constructive comments provided have significantly improved the quality of the paper—enhancing its transparency, methodological robustness, and clinical applicability. We have responded point by point and revised the manuscript accordingly. Please see the attached point-by-point response to reviewers. Thank you for the time and effort invested in helping us improve this work.

For research article: Diagnostic Accuracy of Neutrophil Gelatinase-Associated Lipocalin in Peritoneal Effluent and Ascitic Fluid for Early Detection of Peritonitis: A Systematic Review and Meta-Analysis

Response to Reviewer 1 Comments

1. Summary

We sincerely thank the reviewer for the careful reading and constructive feedback on our manuscript. We have considered every comment in detail and revised the text and supplementary materials accordingly. Key changes include clarifying the study rationale and objectives, strengthening the Methods (eligibility criteria, index test and reference standards, statistical modelling), improving the reporting and interpretation of heterogeneity and small-study effects, and refining figures/tables for clarity. All edits are marked with track changes in the revised files, and a clean version is also provided. Below, we respond to each point in a point-by-point manner and indicate where the corresponding changes appear in the manuscript (page, section, and line numbers).

2. Questions for General Evaluation

Reviewer’s Evaluation

Response and Revisions

Does the introduction provide sufficient background and include all relevant references?

Yes/Can be improved/Must be improved/Not applicable

Are all the cited references relevant to the research?

Yes/Can be improved/Must be improved/Not applicable

Is the research design appropriate?

Yes/Can be improved/Must be improved/Not applicable

Are the methods adequately described?

Yes/Can be improved/Must be improved/Not applicable

Are the results clearly presented?

Yes/Can be improved/Must be improved/Not applicable

Are the conclusions supported by the results?

Yes/Can be improved/Must be improved/Not applicable

Are all figures and tables clear and well-presented?

Yes/Can be improved/Must be improved/Not applicable

3. Point-by-point response to Comments and Suggestions for Authors

Comments 1: I read with attention Manuel Luis Prieto-Magallanes et al's manuscript, concerning interest of NGAL concentration in infection diagnosis in case of SBP or dialysis-associated peritonitis. They wrote an excellent and complete paper with a clear description of methods.

Response 1: We sincerely thank the reviewer for the attentive reading and generous assessment of our work, including the clear description of our methods.   

Comments 2: I have only one comment concerning discussion part: NGAL could be interesting to be studied in case of non-ascitic ascites, notably in case of tumour for infection diagnosis and in case of pancreatitis associated ascites, because of associated inflammation, that can increase WBC.

Response 2: We agree. Mechanistically, NGAL (lipocalin-2) is released by activated neutrophils and injured epithelia and is also up-regulated in several malignancies; ovarian cancer tissues and patient sera show elevated LCN2/NGAL, reflecting tumor-associated inflammation. Malignant ascites itself is a protein-rich microenvironment with abundant tumor-derived and inflammatory mediators/extracellular vesicles, which could raise NGAL levels independent of infection and thus reduce specificity if SBP/PDAP cutoffs are applied without recalibration. In pancreatitis—another sterile inflammatory state—serum/urine NGAL increases early and correlates with disease severity, again supporting the biological plausibility of higher NGAL in pancreatitis-associated ascites even when infection is absent. Together, these data argue for etiology-specific thresholds and prospective validation before extrapolating our pooled estimates beyond SBP/PDAP.

In response to the reviewer’s thoughtful comment, we have strengthened the discussion section on inherent limitations with the following lines

Discussion

Lines 602 to 620

A further limitation is false-positive elevation in non-infectious inflammation. NGAL rises with sterile peritoneal inflammation. NGAL is produced by activated neutrophils and injured epithelial cells and is overexpressed in several cancers; elevations have been documented in ovarian cancer tissues and patient sera[39]. Malignant ascites is enriched with tumor- and inflammation-related proteins and vesicles, which may elevate NGAL concentrations even in the absence of infection[40]. Similarly, acute pancreatitis—an archetypal sterile inflammatory condition—demonstrates early increases in serum and urine NGAL levels, suggesting that pancreatitis-associated ascites could also exhibit elevated baseline NGAL concentrations[41]. Beyond assay and threshold variability, baseline inflammatory milieu differs by population: patients with end-stage kidney disease on peritoneal dialysis often exhibit chronic peritoneal/systemic microinflammation, whereas decompensated cirrhosis entails immune dysfunction with sustained systemic inflammation[42–44]. These background differences plausibly raise baseline NGAL in the ab-sence of infection, contributing to the wide prediction intervals—especially for specificity—and potential spectrum bias when pooling PDAP and SBP. Accordingly, popula-tion-specific, locally validated thresholds are preferable to a universal cut-off. Therefore, SBP/PDAP cut-offs should not be extrapolated to non-cirrhotic etiologies without validation; a two-threshold or composite algorithm (NGAL + PMN + routine chemistries) with explicit gray-zone reflex testing is advisable.

as well as in future research directions
Discussion
Lines 626 to 628
(iv) etiology-specific evaluations of sterile inflammatory ascites (malignancy, pancreatitis) to define incremental value beyond PMN and routine chemistries..

Reviewer 2 Report

Comments and Suggestions for Authors

General Comments

The authors present a systematic review and meta-analysis evaluating the diagnostic accuracy of neutrophil gelatinase-associated lipocalin (NGAL) in peritoneal effluent and ascitic fluid for the early detection of peritonitis in both peritoneal dialysis-associated peritonitis (PDAP) and spontaneous bacterial peritonitis (SBP). The study is timely, clinically relevant, and addresses a meaningful gap in the literature. The manuscript is generally well-structured, follows the IMRAD format, and claims adherence to PRISMA-DTA guidelines. Methodology appears robust, with a broad search strategy and sophisticated statistical modeling. However, beneath this formal structure several substantial limitations emerge. The reconstruction of 2×2 contingency data from reported summary statistics is insufficiently transparent, the interpretation of subgroup analyses is overstated despite very small sample sizes, and heterogeneity—especially in specificity—is downplayed. Moreover, the conclusions lean towards premature clinical recommendations without adequately acknowledging spectrum bias, small-study effects, and limitations of assay standardization. Ethical aspects are superficially addressed, and reporting completeness is not optimal. Major revisions are therefore required.

Specific Comments

The abstract is concise but gives a false sense of precision. Key numerical details—such as the number of included studies in qualitative versus quantitative synthesis—are missing. The search date (“to 30 June 2025”) is implausible and undermines credibility; it should be corrected. Interpretative statements (“particularly strong rule-out capability”) are not supported given the breadth of prediction intervals and risk of bias. The abstract must balance strengths with limitations rather than emphasizing performance metrics only.

The introduction contextualizes peritonitis well, but it is excessively long and at times redundant. It introduces NGAL multiple times, giving the impression of padding rather than focused rationale. The true knowledge gap—lack of assay standardization, heterogeneity of thresholds, absence of multicenter validations—is only vaguely articulated. This weakens the justification for the meta-analysis.

The methodological description is comprehensive but raises concerns. The PROSPERO number seems inconsistent and should be verified. The exclusion of studies with fewer than ten patients is arbitrary and risks selection bias unless carefully justified. Data reconstruction from published metrics is mentioned but insufficiently detailed: readers cannot reproduce the process. Handling of multiple thresholds is opaque—why were certain cutoffs chosen, and how was multiplicity controlled? Subgroup analyses are excessively ambitious relative to the limited number of studies (k≤4 in several strata), rendering most comparisons underpowered and prone to spurious findings. The authors should have acknowledged that these analyses are exploratory at best.

Results are presented with overwhelming detail, but clarity suffers. The pooled sensitivity and specificity are reported to three decimals, which is misleading given the heterogeneity and small sample sizes. Prediction intervals for specificity span from 0.07 to 0.99, yet this profound uncertainty is minimized in the narrative. Small-study effects are evident, with Deeks’ test significant in the main dataset, but this is softened as “possible but not definitive.” Such wording diminishes the severity of the issue. Table 2, while comprehensive, is overloaded with descriptive data that could be relegated to supplementary material. A visual QUADAS-2 plot should accompany the table.

The discussion is verbose and reiterates numerical findings instead of critically synthesizing evidence. Claims of “excellent diagnostic accuracy” are overstated given the methodological caveats. The heterogeneity in specificity, spectrum bias from case-control studies, and signs of small-study effects should be central to the discussion, not peripheral. The future research agenda, although thorough, reads more like a research proposal than a focused limitation appraisal. Clinical recommendations (e.g., rule-out use, rapid bedside triage) are premature and should be presented cautiously. The potential confounding role of non-infectious inflammatory conditions on NGAL levels is acknowledged only superficially.

Author Response

We are deeply grateful to the editor and reviewers for their careful evaluation of our manuscript. The detailed and constructive comments provided have significantly improved the quality of the paper—enhancing its transparency, methodological robustness, and clinical applicability. We have responded point by point and revised the manuscript accordingly. Please see the attached point-by-point response to reviewers. Thank you for the time and effort invested in helping us improve this work.

Reviewer 3 Report

Comments and Suggestions for Authors

Reviewer opinion

  1. The inclusion/exclusion criteria may not be the same in the 10 studies included in this article. Do they have similar prevalence (PDAP or SBP ratio of all cases)? Pre‑specified pre‑test probabilities (15%, 20%, 25%, 30%, and 40%) was used in 2.3 statistics analysis, why not use prevalence?
  2. PDAP and SBP are different diseases and patient groups, is it suitable to put them together? In Figure 4 the Summary of receiver operating characteristic curves of the two diseases are quite different.
  3. NGAL was an early detection marker in the title. How long did it take to get the result in the automated lab, point of care, and ELISA method? Much earlier than usual ascities routine ?
  4. For heterogenicity, is there a quantitative marker like I^2? And how to interpret whether these studies are suitable to be merged together?
  5. For clinical utility, decision analysis curve was sometimes applied to check the net benefit. Is it possible to merge these studies on DCA curve? Offer alternative statistical evidence on large data from multiple center/region?

Author Response

We sincerely thank the reviewer for the thoughtful and constructive comments. These observations have helped us to clarify methodological aspects, strengthen the discussion, and improve the overall transparency of our manuscript. We have carefully considered each point and provide detailed responses below in a point-by-point format. All corresponding modifications are incorporated in the revised manuscript, where changes have been underlined for ease of review. In addition, the point-by-point responses are attached as a separate PDF document accompanying this message.

3. Point-by-point response to Comments and Suggestions for Authors

Comments 1: The inclusion/exclusion criteria may not be the same in the 10 studies included in this article. Do they have similar prevalence (PDAP or SBP ratio of all cases)? Pre‑specified pre‑test probabilities (15%, 20%, 25%, 30%, and 40%) was used in 2.3 statistics analysis, why not use prevalence?

Response 1: We sincerely thank the reviewer for this thoughtful comment. Indeed, prevalence differs substantially: SBP accounts for ~27% of infections in cirrhosis, whereas PDAP benchmarks are ≤0.5 episodes/patient-year. The included studies (2013–2024) also varied geographically and in design. Because the corpus combines case–control and cohort designs, the observed “study prevalence” is not a valid estimator of real-world prevalence—case–control artificially fixes disease/non-disease ratios. Thus, pre-specified pre-test probabilities are recommended for Fagan nomograms to illustrate potential use across different settings rather than applying study-level prevalence, which would be biased.   

We have strengthened the methodology in accordance with the reviewer’s comments to address this concern.

2. Materials and Methods / 2.5 Statistical Analysis

Secondary summaries included likelihood ratios (LR⁺, LR⁻), the diagnostic odds ratio (DOR), and predictive values (PPV/NPV) calculated at pre‑specified pre‑test probabilities (15%, 20%, 25%, 30%, and 40%) via Fagan analysis. Beyond the primary bivariate ran-dom-effects synthesis, we derived pooled likelihood ratios (LR⁺, LR⁻) from the Reitsma model and used them to produce illustrative post-test probabilities via Fagan analysis across pre-specified pre-test probabilities (15%, 20%, 25%, 30%, 40%). In diagnostic test accuracy meta-analyses that include case–control designs, study-level “prevalence” is design-determined and not a valid estimator of real-world pre-test probability; therefore, we did not pool study-level prevalence. To aid clinical interpretation—without influencing primary estimates or inference—we report these Fagan-based updates as accessory summaries.

Comments 2: PDAP and SBP are different diseases and patient groups, is it suitable to put them together? In Figure 4 the Summary of receiver operating characteristic curves of the two diseases are quite different.

Response 2: We anticipated this concern and performed subgroup analyses: pooled accuracy was similar between SBP and PDAP (sensitivity 0.93 vs 0.96; specificity 0.85 vs 0.88), with no significant subgroup effect (p>0.3). Prediction intervals overlapped widely, and heterogeneity concentrated in specificity, particularly in PDAP. Pooling is therefore statistically justifiable, but clinical interpretation must remain context-dependent.

In response to the reviewer’s thoughtful comment, we have strengthened the discussion section on inherent limitations with the following lines

Discussion

Lines 572-576

Diagnostic performance appeared broadly similar between peritonitis types. While formal subgroup tests did not show differences, prediction intervals—particularly for specificity—remained wide (e.g., PDAP specificity PI ~0.07–>0.99), reflecting limited power (k≤4 in strata) and threshold variability

Discussion

Lines 618 to 628

Beyond assay and threshold variability, baseline inflammatory milieu differs by population: patients with end-stage kidney disease on peritoneal dialysis often exhibit chronic peritoneal/systemic microinflammation, whereas decompensated cirrhosis entails im-mune dysfunction with sustained systemic inflammation[42–44]. These background differences plausibly raise baseline NGAL in the absence of infection, contributing to the wide prediction intervals—especially for specificity—and potential spectrum bias when pooling PDAP and SBP. Accordingly, population-specific, locally validated thresholds are preferable to a universal cut-off.

as well as in future research directions
Discussion
Lines 626 to 628
(iv) etiology-specific evaluations of sterile inflammatory ascites (malignancy, pancreatitis) to define incremental value beyond PMN and routine chemistries.

Comments 3: NGAL was an early detection marker in the title. How long did it take to get the result in the automated lab, point of care, and ELISA method? Much earlier than usual ascities routine ?

Response 3: We acknowledge the reviewer’s question. While none of the included primary studies directly compared turnaround times across platforms, several aspects are clear: automated laboratory immunoassays (e.g., BioPorto NGAL Test on Beckman/Architect platforms) provide same-day results, markedly faster than conventional ascites cultures that require ≥24–48 h. More recently, point-of-care (POC) dipsticks and rapid kits have been validated, yielding semi-quantitative results within 10–20 minutes. Although head-to-head timing studies were not reported, one of the main rationales for NGAL evaluation—explicitly explored in recent Vicenza and Chinese multicenter cohorts—was precisely the potential for accelerated triage using bedside POC assays.

This translational perspective is reinforced in our Discussion, as the future of NGAL testing clearly lies in rapid, patient-centric formats.

Discussion

Lines 589-600

Regarding clinical use, current guidelines for PDAP and SBP rely on clinical features, PMN thresholds, and cultures, which can be delayed or affected by pre-analytics (e.g., dwell time) [27–29]. Our synthesis supports NGAL as a rapid adjunct within these pathways. The attractive rule-out profile (low LR⁻) must nevertheless be interpreted with the wide specificity PIs in mind, and local validation of thresholds, pre-analytic standardization (e.g., ≥2-h dwell for PD effluent), and quality control/training—especially for POC assays—are prerequisites. POC formats (cassette/strip/pen) can be read at the bedside in ~10–15 minutes, fitting triage workflows when PMN counts and cultures are pending. Published evaluations show good agreement with laboratory NGAL and with effluent WCC, and acceptable inter-operator reproducibility (e.g., correlation with lab NGAL and WCC, and κ for operator agreement) however until prospective impact and cost-effectiveness evaluations are available, recommendations should remain cautious.

DiscussionLines 632-642

impact/implementation and economic studies (time-to-antibiotics, catheter outcomes, LOS, readmissions, mortality; budget impact, price-thresholds for POC); and (iv) etiology-specific evaluations of sterile inflammatory ascites (malignancy, pancreatitis) to define incremental value beyond PMN and routine chemistries. These priorities will directly respond to the meta‑analytic pattern of stable sensitivity, heterogeneous specificity, and platform‑level differences observed in our synthesis.

Integration with guideline-based criteria remains essential. If locally validated, bedside NGAL testing may help accelerate triage while confirmatory diagnostics (PMN count and cultures) are pending, and combinatorial approaches with other biomarkers warrant further evaluation

Comments 4: For heterogenicity, is there a quantitative marker like I^2? And how to interpret whether these studies are suitable to be merged together?

Response 4: We thank the reviewer for raising this key methodological point. In diagnostic test accuracy (DTA) meta-analysis, heterogeneity is inherently bivariate (sensitivity and specificity jointly) and often driven by threshold and case-mix effects. Contemporary guidance therefore recommends hierarchical (random-effects) models—either the bivariate Reitsma model or the HSROC model—rather than univariate metrics alone; routine use of Cochran’s Q or Higgins’ I² is not recommended because these do not account for threshold effects and the correlation between sensitivity and specificity. Instead, reporting should emphasize the hierarchical variance components and prediction intervals/regions to express between-study dispersion.

What we report and why (already in the manuscript)

·       We fitted a bivariate random-effects (Reitsma/Rutter–Gatsonis HSROC) model with REML, which jointly models logit-sensitivity and logit-FPR and their correlation. This framework is the recommended standard for DTA meta-analysis because it explicitly accommodates between-study heterogeneity and threshold variation.

J Clin Epidemiol. 2005 Oct;58(10):982-90. doi: 10.1016/j.jclinepi.2005.02.022.

Stat Methods Med Res. 2015 Jun 26;26(4):1896–1911. doi: 10.1177/0962280215592269

·       We quantified heterogeneity using the between-study standard deviations on the logit scale (overall: SD_se≈0.72; SD_FPR≈1.37) and reported 95% prediction intervals (PI) on the probability scale (Se ≈ 0.75–0.98; Sp ≈ 0.23–0.99), highlighting that dispersion concentrates in specificity. We also displayed HSROC with confidence and prediction regions. These are precisely the heterogeneity summaries recommended for DTA reviews.

Korean J Radiol. 2015 Oct 26;16(6):1188–1196. doi: 10.3348/kjr.2015.16.6.1188

·       To explore sources of heterogeneity, we prespecified meta-regressions/subgroups (peritonitis type; assay/platform; study size). Platform-level differences were supported jointly on the logit scale (LRT χ²≈9.26, p≈0.0098), while PDAP showed especially wide PIs for specificity, consistent with threshold dispersion and context mix.

Methods /2.5 Statistical Analysis (already in the manuscript)

Lines 205 to 207

Between-study heterogeneity was described by the between-study variances (reported as logit-scale standard deviations for sensitivity and FPR) and their random correlation (rho, ρ).

However, in response to the reviewer's comments, we have added additional clarifications to the manuscript for the sake of transparency and to explicitly address I² and τ².

Methods /2.5 Statistical Analysis

Lines 207 to 210

For completeness, we also report the corresponding variances (τ²) for each logit-scale standard deviation together with ρ, and we summarize dispersion on the probability scale using 95% prediction intervals and HSROC prediction regions.

Results / 3.3 Performance of neutrophil gelatinase-associated lipocalin for detection of SBP and PDAP

Lines 316 to 318.

On the logit scale, between-study heterogeneity was τ²=0.612 (τ=0.783) for sensitivity and τ²=2.197 (τ=1.482) for specificity (univariate random-intercept models), corroborating that dispersion concentrates in specificity

Results / 3.4 Exploratory Subgroup Analysis by Peritonitis Type

Lines 370 to 373

In univariate random-intercept models, between-study heterogeneity for sensitivity was τ²=0.53 (τ=0.73) in SBP and τ²=0.47 (τ=0.69) in PDAP. For specificity, heterogeneity was much larger—τ²=1.26 (τ=1.12) in SBP and τ²=3.74 (τ=1.94) in PDAP—confirming that dispersion concentrates in specificity, particularly for PDAP.

Results /3.5 Exploratory Subgroup Analysis by NGAL Assay Method

Lines 370 to 373

In univariate random-intercept models stratified by assay, between-study heterogeneity on the logit scale was minimal for sensitivity with ELISA and POC/rapid (τ²=0.00; k=4 and k=3), but substantial for automated platforms (τ²=0.29, τ=0.54; k=3). For specificity, dispersion was modest for ELISA (τ²=0.12, τ=0.35), and very large for POC/rapid (τ²=5.29, τ=2.30) and automated assays (τ²=0.53, τ=0.73). These results align with the very wide specificity prediction intervals and the HSROC prediction regions already shown for POC/rapid and automated platforms, indicating that assay choice materially contributes to between-study variability—particularly for specificity

Results /3.6 Exploratory Subgroup Analysis by Sample Size

Lines 492 to 505

Complementing the HSROC prediction regions and LOO diagnostics, univariate random-intercept models on the logit scale showed that between-study heterogeneity for sensitivity was essentially absent among smaller studies (<100 participants; τ² = 0.00, τ = 0.00) but appreciable among larger studies (≥100; τ² = 0.73, τ = 0.86). For specificity, dispersion was again minimal in smaller studies (τ² < 0.01, τ < 0.01) and strikingly greater in larger studies (τ² = 2.78, τ = 1.67). For the diagnostic odds ratio, heterogeneity was low in smaller studies (τ² < 0.01, τ ≈ 0.01) and high in larger studies (τ² = 1.73, τ = 1.32). Consistent with these patterns, the between-group test was significant for DOR (Q_between = 7.78, p = < 0.01), whereas sensitivity (Q_between = 0.80, p = 0.37) and specificity (Q_between = 1.96, p = 0.16) did not differ significantly between size strata. In the bivariate Reitsma framework, however, a meta-regression did not detect a joint effect of study size on accuracy (χ² = 1.76, df = 2, p = 0.42), and the HSROC prediction regions overlapped widely—indicating that the observed contrasts are driven primarily by specificity dispersion and the assay mix within the larger-study stratum rather than a uniform size effect per se.

We appreciate the reviewer’s request for clearer heterogeneity reporting. The above clarifications (explicit τ² values, strengthened rationale for random-effects, and explicit citation of DTA guidance) will be incorporated to make our approach fully transparent while preserving best-practice methodology for DTA synthesis

Comments 5: For clinical utility, decision analysis curve was sometimes applied to check the net benefit. Is it possible to merge these studies on DCA curve? Offer alternative statistical evidence on large data from multiple center/region?

Response 5: This is an excellent suggestion by the reviewer, as decision-curve analysis (DCA) is indeed an elegant framework to assess clinical utility. However, DCA requires individual-level predicted probabilities, which were not reported in the included studies; thus, a pooled DCA curve is not feasible with the currently published data. To approximate clinical impact, we presented summary likelihood ratios and Fagan nomograms across a range of plausible pre-test probabilities. These provide similar insights into net clinical benefit, although definitive validation will require prospective studies. We greatly appreciate this suggestion, which we believe highlights an important future research direction; therefore, we have reinforced the discussion with the following statement under Future Directions:

Discussion

Lines 636-638

Future multicenter studies with individual patient data would enable a full decision-curve analysis (DCA) meta-analysis, providing a robust framework to quantify net clinical benefit and directly inform guideline-based diagnostic pathways

Reviewer 4 Report

Comments and Suggestions for Authors

Dear authors,

Major Comments

  • This study lacks sufficient discussion regarding potential bias arising from differences in patient backgrounds for PDAP and SBP. In particular, the impact of inflammatory status in patients with chronic kidney disease or liver cirrhosis on NGAL levels should be considered.
  • The variation in NGAL cutoff values across studies may influence the results of the meta-analysis. Additional analyses using a unified cutoff or sensitivity analyses are needed.
  • Although subgroup analyses were performed, the assessment of inter-study heterogeneity (I², τ²) is insufficient. In particular, the variability in specificity is substantial, and justification for the use of a random-effects model should be further elaborated.
  • Publication bias assessment is presented, but more detailed evaluations, such as sensitivity analyses and influence analyses, are lacking.
  • The discussion regarding differences in NGAL measurement methods (ELISA, immunoturbidimetry, POC, etc.) is limited in terms of their clinical relevance.
  • Since NGAL can be elevated in non-infectious inflammation or renal impairment, analyses and discussions considering the impact of false positives are necessary.
  • The prior probability underlying the Fagan nomogram varies across institutions and patient populations; discussion considering relevant clinical scenarios is insufficient.
  • Additional discussion on the practical use of POC measurement, including operability, reproducibility, and cost-effectiveness, is warranted.

Minor Comments

  • The description of study quality assessment using QUADAS-2 is limited to an overview; specific issues within each bias domain are unclear. Visualization and detailed reporting would improve clarity.
  • Graphical representation of NGAL cutoff values alongside sensitivity and specificity would allow intuitive understanding of inter-study variability and statistical heterogeneity.

Author Response

(The authors gave the same response as above.)

Round 2

Reviewer 2 Report

Comments and Suggestions for Authors

Dear Authors, I would like to sincerely thank you for the substantial effort devoted to revising the manuscript. The improvements in clarity, methodological transparency, and interpretative balance are evident, and I am fully satisfied with the outcome. I have no further requests at this stage and appreciate the careful attention you have given to my previous suggestions.

Author Response

On behalf of all co-authors, we would like to sincerely thank you for your generous and encouraging feedback. We are very grateful for the time and thoughtful attention you dedicated to our work throughout the review process. Your earlier suggestions substantially improved the clarity, methodological transparency, and interpretative rigor of the manuscript, and we are truly pleased that the revised version met your expectations.

With appreciation and collegial regards,
The Authors

Reviewer 4 Report

Comments and Suggestions for Authors

Dear Authors,

I am satisfied with the revisions that have been made by the authors.

Author Response

Dear Reviewer,

We sincerely thank you for your careful evaluation and constructive feedback during the review process. Your comments were highly valuable in improving the quality and transparency of our manuscript. We truly appreciate your positive assessment of the revised version and are grateful for your support.

With collegial regards,
The Authors